# Theoretical Study of Structure and Photophysics of Homologous Series of Bis(arylydene)cycloalkanones

**DOI:** 10.3390/ijms241713362

**Published:** 2023-08-29

**Authors:** Roman O. Starostin, Alexandra Ya. Freidzon, Sergey P. Gromov

**Affiliations:** 1FSRC “Crystallography and Photonics”, Photochemistry Center of RAS, Russian Academy of Sciences, Novatorov Str. 7A-1, Moscow 119421, Russia; star.roman-96@yandex.ru (R.O.S.); spgromov@mail.ru (S.P.G.); 2Department of Chemistry, M. V. Lomonosov Moscow State University, Leninskie Gory 1-3, Moscow 119991, Russia; 3Institute of Nanoengineering in Electronics, Spintronics and Photonics, National Research Nuclear University MEPhI, Kashirskoye Shosse, 31, Moscow 115409, Russia; 4Faculty of Chemistry, Molecular Chemistry and Materials Science, Weizmann Institute of Science, 234 Herzl Street, P.O. Box 26, Rehovot 7610001, Israel

**Keywords:** bis(arylydene)cycloalkanone dyes, photophysical properties, photochemical properties, absorption, luminescence, trans–cis isomerization, quantum chemistry, density functional theory, potential energy surface, conical intersection

## Abstract

Photophysical properties of a series of bis(arylydene)cycloalkanone dyes with various donor substituents are studied using quantum chemistry. Their capacity for luminescence and nonradiative relaxation through *trans–cis* isomerization is related to their structure, in particular, to the donor capacity of the substituents and the degree of conjugation due to the central cycloalkanone moiety. It is shown that cyclohexanone central moiety introduces distortions and disrupts the conjugation, thus leading to a nonmonotonic change in their properties. The increasing donor capacity of the substituents causes increase in the HOMO energy (rise in the oxidation potential) and decrease in the HOMO–LUMO gap, which results in the red shift of the absorption spectra. The ability of the excited dye to relax through fluorescence or through *trans–cis* isomerization is governed by the height of the barrier between the Franck–Condon and S1–S0 conical intersection regions on the potential energy surface of the lowest π-π* excited state. This barrier also correlates with the donor capacity of the substituents and the degree of conjugation between the central and donor moieties. The calculated fluorescence and *trans–cis* isomerization rates are in good agreement with the observed fluorescence quantum yields.

## 1. Introduction

Bis(arylidene)cycloalkanones (Figure 1), also known as symmetric cross-conjugated dienones, ketocyanine dyes, or diarylidene ketone derivatives, are D-π-A-π-D dyes with interesting photochemistry and photophysics. The electrochemistry, photophysics, and photochemistry of bis(arylidene)cycloalkanone series have been studied extensively [1,2,3]. The compounds were found to exhibit efficient *trans–cis* photoisomerization reactions. This property makes bis(arylidene)cycloalkanones useful in the development of photoresponsive materials and devices, such as molecular switches and optical data storage. Bis(arylidene)cycloalkanone compounds display solvatochromic properties [4,5,6] and can be applied, for example, for determining the polarity of a medium [7,8,9] or as pH sensors [10]. Bis(arylidene)cycloalkanones modified with ionophoric fragments [11,12,13] form a basis for photocontrolled supramolecular devices. Nanoparticles are used to modify the spectra of ketocyanine dyes [14].

Bis(arylidene)cycloalkanone compounds were found to exhibit strong two-photon absorption properties, which could be useful for two-photon microscopy [15] and photodynamic therapy [16,17,18]. The studies found that the photophysical properties of the compounds were dependent on the size of the alicyclic ring, with larger rings leading to lower fluorescence quantum yields and shorter excited-state lifetimes [13]. Ref. [19] discusses the recent advances on benzylidene cyclopentanones as photosensitizers for two-photon polymerization. The study focuses on the symmetrically substituted benzylidene cyclopentanones and their ability to initiate polymerization. Ref. [20] discusses the molecular structure and vibrational spectra of 2,6-bis(benzylidene)cyclohexanone molecule. The paper presents the near-infrared Fourier transform (NIR-FT) Raman and Fourier transform infrared (FT-IR) spectra of 2,6-bis(benzylidene)cyclohexanone molecule along with the density functional calculations. Numerous papers [21,22,23] discuss the photoprocesses, kinetics, and photoproducts of bis(arylidene)cycloalkanones and their derivatives with electron-donating substituents. The studies analyze the spectral, luminescent, and time-resolved properties of the compounds and their derivatives. Ref. [24] theoretically studies the impact of conjugation length on the electronic structure, small energy optical absorption and third-order polarizabilities of symmetric ketocyanine dyes. Ref. [25] demonstrates this by experiment.

The electrochemistry and positions of the absorption and emission bands are immediately related to the electronic structure of the molecules in study, while the luminescence quantum yields and excitation decay kinetics are governed by the potential energy surfaces. The goal of this paper is to provide a quantum chemical interpretation of the photochemical and photophysical properties of the cycloalkanone series as a function of their molecular structure and important features of their potential energy surfaces. This study will help one gain deeper understanding of the structure–property relationships of bis(arylidene)cycloalkanone derivatives and facilitate their targeted molecular design.

## 2. Results

We consider a series of bis(benzylidene)cycloalkanone derivatives (Figure 2) with electron-donating groups on the phenyl rings. In polar solvents, such as acetonitrile, the global energy minimum corresponds to the sterically unhindered (*E*,*E*) isomers, while the gas phase structure, as predicted by the calculations, is (*Z*,*Z*) form stabilized by intramolecular C-H…O hydrogen bonds. Therefore, all the calculations were performed taking solvent into account within polarizable continuum model [26].

### 2.1. Ground State Structures and Molecular Orbitals

The calculations demonstrated that the fractions of (*E*,*Z*) and (*Z*,*Z*) isomers is negligible in the ground state. This agrees with the experiment [1,2,3,22], where no electronic transitions corresponding to (*E*,*Z*) and (*Z*,*Z*) isomers were observed in the absorption spectra. The ground-state rotation barriers are ~50–60 kcal/mol, which indicates that *trans–cis* isomerization in the ground state is impossible at any reasonable temperature.

The depths of the local minima associated with the rotation of the phenyl rings (shown by arrows in Figure 2) differ by a fraction of kilocalorie, and these minima are separated by low barriers, which can easily be overcome at room temperature, ensuring slightly hindered rotation of these groups. The 1H NMR experiments indicate that dyes **(1–3)c** exist as mixtures of rotamers [1,2,3].

The planarity of the molecule reflects their degree of conjugation. In bis(benzylidene)cycloalkanones under study, the central part is rather rigid, but rotation is possible around formally single bonds shown in Figure 2 by arrows. Table 1 shows the corresponding torsion angles characterizing the effect caused by the central alicyclic fragment on the conjugated chromophore chain.

One can observe that cyclobutanone central moiety favors perfectly conjugated chromophore chain, and the distortion introduced by cyclopentanone is minor, while that introduced by cyclohexanone is noticeable. Donor substituents favor the conjugation diminishing the torsion from 7° in **2a** to almost 0° in **2d,e**. The distortion caused by cyclohexanone is so large that even donor substituents reduce it from 30° in **3a** to only 21° in **3e**. The same trend is observed in the available crystal structures: 0–2° in cyclobutanones, 1–8° in cyclopentanones, and 9–39° in cyclohexanones.

The chemical shifts (Appendix A) show the following trend with the size of the central alicycle: the aliphatic protons shift towards strong fields, while methine protons shift towards weak fields. This trend is observed both in the calculated and experimental chemical shifts.

The donor capacity of the substituents is reflected by the Mulliken charges induced by the substituents on the phenyl rings (Table 2). In more conjugated systems, the electron density can flow freely from the donors to the acceptor, thus creating positive charge on the donors and negative on the acceptor. Due to the distorted conjugation, the electron density remains on the donors, and the charge on the donor fragment slightly decreases in cyclohexanone relative to cyclobutanone and cyclopentanone. As for the donor capacity of the substituents, the charges suggest the following order: 4-H < 4-OMe ~ 3,4-OMe ~ 4-SMe < 4-NEt2.

Another way of characterizing donor capacity of the substituents is their effect on the frontier orbital energies, mainly HOMO. Figure 3 shows that the trend observed in the Mulliken charges is more evident in the HOMO energy: 4-H < 4-OMe < 3,4-OMe ~ 4-SMe < 4-NEt2. The LUMO energy is mostly affected by the acceptor fragment, but the effect is significantly less evident. The HOMO–LUMO gap, which correlates with the absorption spectra, decreases in the series: 4-H > 4-OMe > 3,4-OMe~4-SMe > 4-NEt2. Similar correlations were found in [27].

The HOMO–LUMO gap in cyclohexanone derivatives is in general higher than that in cyclobutanones and cyclopentanones owing to the distorted conjugation caused by the cyclohexanone moiety. This results in blue-shifted spectra of the cyclohexanone derivatives. The HOMO and LUMO energies correlate with the oxidation and reduction potentials [1,2,3]. The HOMO–LUMO gaps also correlate with the experimental *E*_ox_–*E*_red_ gaps (refer Appendix A).

In the Hartree–Fock theory, orbital energies of the occupied orbitals are negative for the corresponding ionization potentials (IPs), and orbital energies of unoccupied orbitals are negative for the electron affinities (EAs). This relationship is called Koopmans theorem [28,29]. In the density functional theory, however, this is only valid for the HOMO and first IP. Nevertheless, some correlation can be observed between the LUMO energy and first EA. In addition, HOMO–LUMO gap in DFT correlates with the energy of the first electronic transition. At the same time, IP and EA are related to the oxidation and reduction potentials. Appendix A of Appendix A shows the calculated IPs and EAs compared to the experimental *E*_ox_ and *E*_red_. The correlation is also very good.

In the case of the perfect agreement between the calculated and experimental values, the linear regression coefficient should be close to 1. Our data shows that for the HOMO, IP, and *E*_ox_, this coefficient is more than 0.9, which indicates very good agreement. For the LUMO, EA, and *E*_red_, this coefficient is ~0.7–0.8 with a very high R^2^ value. This is not surprising because the energies of virtual orbitals both in Hartree–Fock and DFT are calculated with less reliability. To some extent, it may be remedied through the use of optimally tuned range-separated functionals [30,31]. The intercept on the linear regressions of *E*_ox_(*E*_red_) vs. HOMO(LUMO) or vs. IP(EA) incorporates all the systematic errors and the absolute potential of the reference electrode.

### 2.2. Absorption and Emission Spectra

The most interesting properties of bis(benzylidene)cycloalkanones are their absorption and emission spectra and their phototransformations. In this section, we consider the energy of the lowest π-π* electronic transition as a function of the donor capacity of the substituent and the size of the central alicycle (refer Appendix A).

The trends in the energy of the lowest π-π* electronic transition (Figure 4) resemble those of the HOMO–LUMO gap because these transitions are mainly contributed by HOMO→LUMO excitation. As before, cyclohexanone dyes have higher excitation energies due to the distorted conjugation.

Our calculations show that in addition to the intense π-π* and dark n-π* electronic transitions, the dyes exhibit one more π-π* electronic transition at ~400 nm resulting from HOMO-1→LUMO transition (Figure 5). The intensity of this transition decreases from cyclobutanone to cyclohexanone (Appendix A). This is supported by the experimental data [1,2,3]: in cyclobutanone **1e** the long-wave absorption band consists of two distinct peaks that cannot be attributed to the vibronic progression, while in **2e** and **3e**, the long-wave absorption band is only broadened. The position of this band changes only slightly, which can be explained by the fact that HOMO-1 is not affected either by the substituents or by the central moiety.

The trends in the order of excited states are shown in Appendix A. The nπ* state is the lowest in the unsubstituted dyes **(1–3)a**, and increases as soon as donor substituents appear in the *para* position. At the same time, the two ππ* states decrease. In **(1–3)b**, the nπ* state lies between the two ππ* states. In the dyes with more donor substituents, the order of states depends on the central alicycle: in the most conjugated **1(c**–**e)**, the nπ* state lies above both ππ* states, while in the other cycloalkanones, the nπ* state lies above both ππ* states only in **(2,3)e**.

Frequently the changes taking place in excited dyes, in particular, the bond length alternation in the excited state, which facilitates *trans–cis* isomerization, are explained in terms of resonance structures (for example, [32]). In most cases, these features are reflected by the nodal structure of HOMO and LUMO. The latter, being populated in the excited state, correlates with the resonance structure with alternated bonds. In the case of benzylidene cycloalkanones, the nodal structure of LUMO also corresponds to the resonance structure with alternated bonds. The bond length in the first ππ* excited state changes in agreement with the nodal structure of LUMO where appearance of the node in LUMO as compared to HOMO results in bond lengthening, while disappearance of the node results in bond shortening. The change is noticeable, especially in the **(1–3)a**, up to 0.03–0.04 A. In general, the largest alternations are in the exocyclic double bonds and in the adjacent single bonds shown by arrows in Figure 2. In some cases, the carbonyl bond also changes.

All cyclobutanone dyes except for **1a** and **1b** are fluorescent, with the quantum yields increasing from **1c** to **1e**. Cyclopentanones **2c–e** are also fluorescent, while in cyclohexanones, only **3e** is fluorescent. For **(1–3)a**, the lack of luminescence can be explained by the fact that their first electronic transition is dark n-π*; therefore, excitation to the bright S2 (ππ*) only causes relaxation to the dark S1(nπ*), which further relaxes nonradiatively through internal conversion or intersystem crossing.

The lack of luminescence of other dyes, whose first electronic transition is bright π-π*, is more difficult to explain. To achieve this, one needs to consider the potential energy surfaces of the ground and first excited states or, more specifically, their cross-sections (profiles) along certain internal coordinates (Figure 6a).

The internal coordinate involved in the relaxation of the S1 (ππ*) excited state of bis(arylidene)cycloalkanones is rotation along a formally double bond. This shape of the potential energy profile implies that the molecule excited from its global (*E*,*E*) minimum to the Franck–Condon region of the S1 state quickly relaxes to its nearest minimum. From this local minimum of the S1 state, the molecule can either emit light or further relax to the region of lower energy (here, it is the region of the S1–S0 conical intersection). If the barrier separating the local minimum from the CI region is not very high, this relaxation can successfully compete with the radiative relaxation. Near the CI point, the molecule quickly relaxes nonradiatively to the S0 state. From this point, the twisted chromophore can relax either back to the (*E*,*E*) region or forward to the (*E*,*Z*) region resulting in *trans–cis* isomerization. Since the S1 state in the local minimum is rather short-lived (either owing to the radiative relaxation or to the structural relaxation to the CI), no equilibrium is possible on the S1 potential energy surface, and excited (*E*,*Z*)* states are not accessible directly from the excited (*E*,*E*)* states. All these processes are schematically shown in Figure 6b.

The right-hand part of the profile in Figure 6a represents the same process, but starting from the (*E*,*Z*) ground state structure. However, since the ground state is dominated by the (*E*,*E*) isomer, the right-hand part of the profile does not seem to be relevant.

Therefore, the luminescence quantum yield is governed by the competition between the radiative relaxation from the (*E*,*E*)* minimum to *trans–cis* photoisomerization. The triplet processes proceed on a microsecond timescale or even slower; therefore, there is no need to include them in the scheme.

Figure 7 shows the trends in the activation energy and CI depth as a function of donor substituents and central ring size. Although the CI depth cannot be a driving force for the *trans–cis* isomerization process, it is indicative of the dye properties. The CI depth decreases and the activation energy increases with the increasing donor capacity of the substituent. The barrier height also increases from cyclohexanone to cyclobutanone. This is in line with the observed trend in the fluorescence quantum yields.

Table 3 shows the radiative lifetimes calculated using the oscillator strengths of the S1–S0 transition, and characteristic *trans–cis* isomerization times calculated using the Arrhenius equation on the S1 potential energy surface of the *trans* isomer. One can observe that emission can proceed on a nanosecond timescale, while *trans–cis* isomerization time can range from fractions of nanosecond to microseconds or longer. Fast isomerization successfully competes with the radiative decay channel, thus causing fluorescence quenching. One can observe that *trans–cis* isomerization in **1(c–e)** is several orders of magnitude slower than in **1(a,b)**, and isomerization of **3b** competes with the radiative relaxation. In **3(c,d)**, relaxation via triplet state adds to the relaxation via *trans–cis* isomerization. The calculated *trans–cis* isomerization times correlate with the observed fluorescence of the cycloalkanones (Appendix A).

## 3. Materials and Methods

The structures and energies of the molecules were calculated using the density functional theory (DFT) with the PBE0 functional and 6-31+G(d,p) basis set of the FireFly program [33], partially based on GAMESS code [34]. The solvent (MeCN) effects were taken into account using the dielectric polarizable continuum model (D-PCM) [26]. Previously [1,2,12], we have shown that for dienones, solvent effects are important to properly reproduce the structures and conformation energies.

The vertical absorption and emission spectra, energy profiles of (*E*,*E*)-(*E*,*Z*) isomerization, and rotation of the aromatic ring around C(β)-C(γ) bond were calculated by the time-dependent DFT (TDDFT) with the same functional, basis set, and solvent model. The vertical absorption spectra were calculated using the TDDFT after DFT optimization of the ground state geometry. A similar method was applied for vertical emission spectra calculations after geometry optimization of the π-π* lowest excited state using the TDDFT and D-PCM. The radiative lifetimes were calculated according to the formula:*k*_r_ = (⅔)*f*_0i_ν^2^_0i_; τ_r_ = 1/*k*_r_
where *f*_0i_ and ν_0i_ are oscillator strength and the frequency of the electronic transition; *k*_r_ is the radiation constant. The isomerization periods were calculated according to the formula:*k*_tc_ = *c*ν_i_·exp(−*E*_Ai_/*RT*); t_tc_ = 1/*k*_tc_
where ν_i_ is the vibrational mode frequency of the *i*th isomer, and *E*_Ai_ is the activation barrier of this isomer.

The main channel of the structural relaxation of the S1 state of (*E*,*E*) bis(benzylidene)cycloalkanones is rotation around a formally double bond leading to *trans–cis* isomerization. To construct the profiles of (*E*,*E*)-(*E*,*Z*) isomerization and an alternative channel, rotation of the aromatic ring around a formally single bond, we used a simple (unrelaxed) scan of the potential energy surface along the corresponding dihedral angles. The energy values correspond to the non-optimized structures obtained by twisting of the initial isomer or rotamer. In the case of (*E*,*E*)-(*E*,*Z*) isomerization, the left-hand rotation barriers were estimated by the energy difference of the maximum on the S1 profile (corresponding to the left transition state) and stable structures of the (*E*,*E*) isomers in the S1 state (left minimum).

We understand that phototransformation, which proceeds via a conical intersection, requires multireference quantum chemistry for an adequate description of the potential energy profiles [35]. Nevertheless, our semi-quantitative description provides insights into the mechanism of phototransformations in organic dyes.

The vertical ionization potentials (IP) and electron affinities (EA) were calculated using restricted-open-shell DFT (RO-DFT) for the corresponding monocation and monoanion of each dye. The functional, basis set, and solvation model were the same.

1H NMR spectra were calculated relative to TMS as a standard using the Priroda program package [36,37] with the PBE functional and triple-ζ quality basis set 3z implemented in the program. The optimized geometries were obtained from the PBE0/6-31+G(d,p)/DPCM calculation.

## 4. Conclusions

Photophysical properties of a series of bis(arylydene)cycloalkanone dyes with various donor substituents are studied using quantum chemistry. Their capacity for luminescence and nonradiative relaxation through *trans–cis* isomerization is related to their structure, in particular, to the donor capacity of the substituents and the degree of conjugation due to the central cycloalkanone moiety. It is shown that cyclohexanone central moiety introduces distortions and disrupts the conjugation, thus leading to a nonmonotonic change in their properties. The donor capacity of the substituents is found to increase in the series 4-H < 4-OMe < 3,4-OMe~4-SMe < 4-NEt2, which causes an increase in the HOMO energy (rise in the oxidation potential) and decrease in the HOMO–LUMO gap (decrease in the excitation energy and a red shift of the absorption spectra). The ability of the excited dye to relax through fluorescence or through the *trans–cis* isomerization is governed by the height of the barrier between the Franck–Condon and S1–S0 conical intersection regions on the potential energy surface of the lowest π-π* excited state. This barrier also correlates with the donor capacity of the substituents and the degree of conjugation between the central and donor moieties. The calculated fluorescence and *trans–cis* isomerization rates are in good agreement with the observed fluorescence quantum yields.

## Figures and Tables

**Figure 1 ijms-24-13362-f001:**
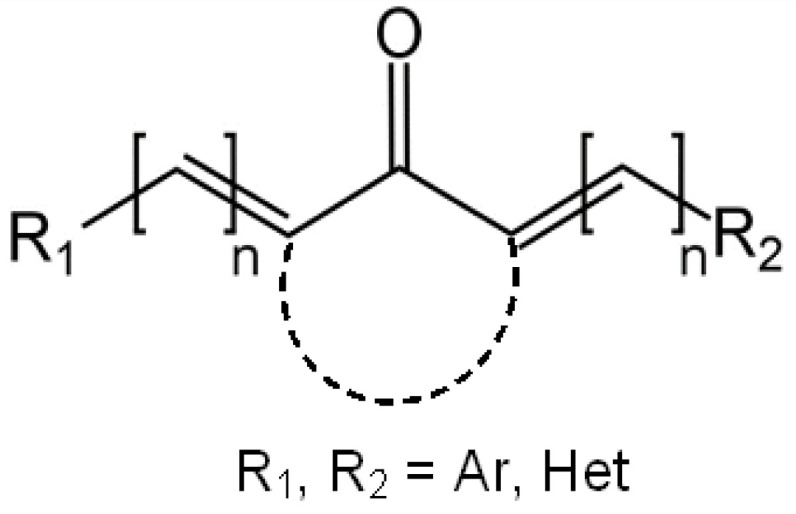
Schematic structure of ketocyanine dyes.

**Figure 2 ijms-24-13362-f002:**
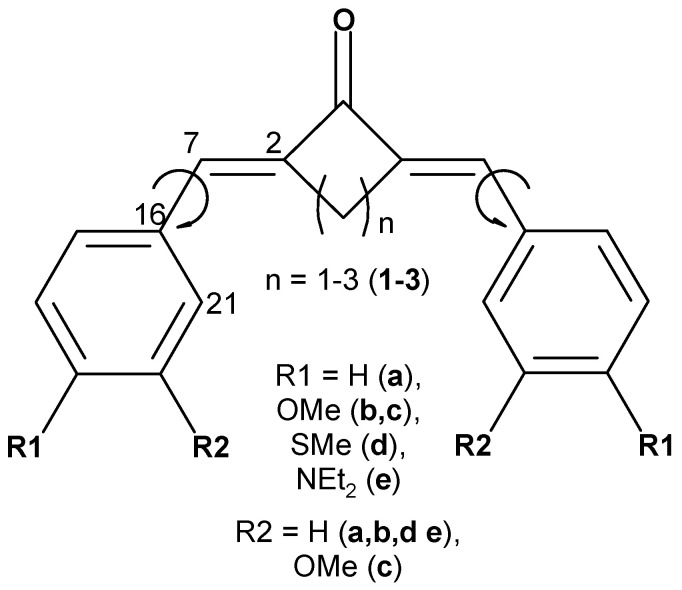
Dyes under study with important torsion angles shown by arrows.

**Figure 3 ijms-24-13362-f003:**
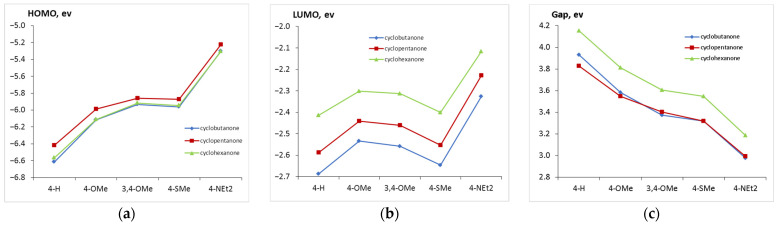
(**a**) HOMO and (**b**) LUMO energies and (**c**) HOMO–LUMO gap as functions of the donor substituents.

**Figure 4 ijms-24-13362-f004:**
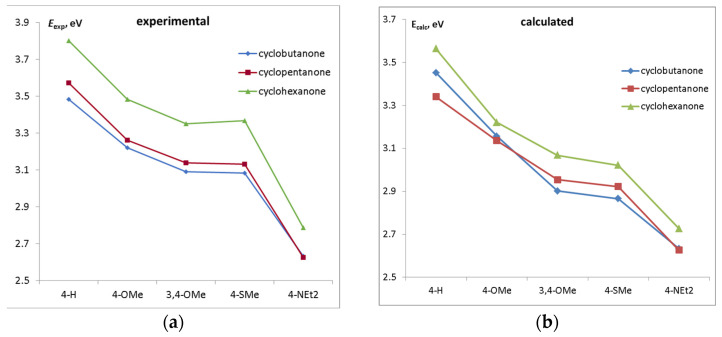
(**a**) Experimental [1,2,3] and (**b**) calculated energy of the lowest π-π* electronic transition as a function of the donor substituent.

**Figure 5 ijms-24-13362-f005:**
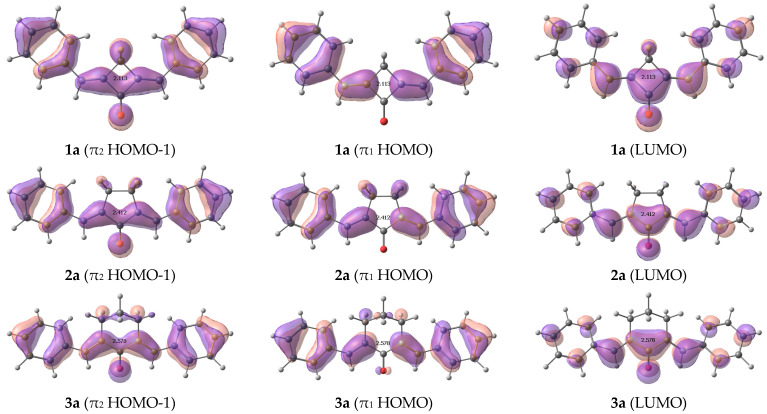
Frontier orbitals of **(1–3)a**.

**Figure 6 ijms-24-13362-f006:**
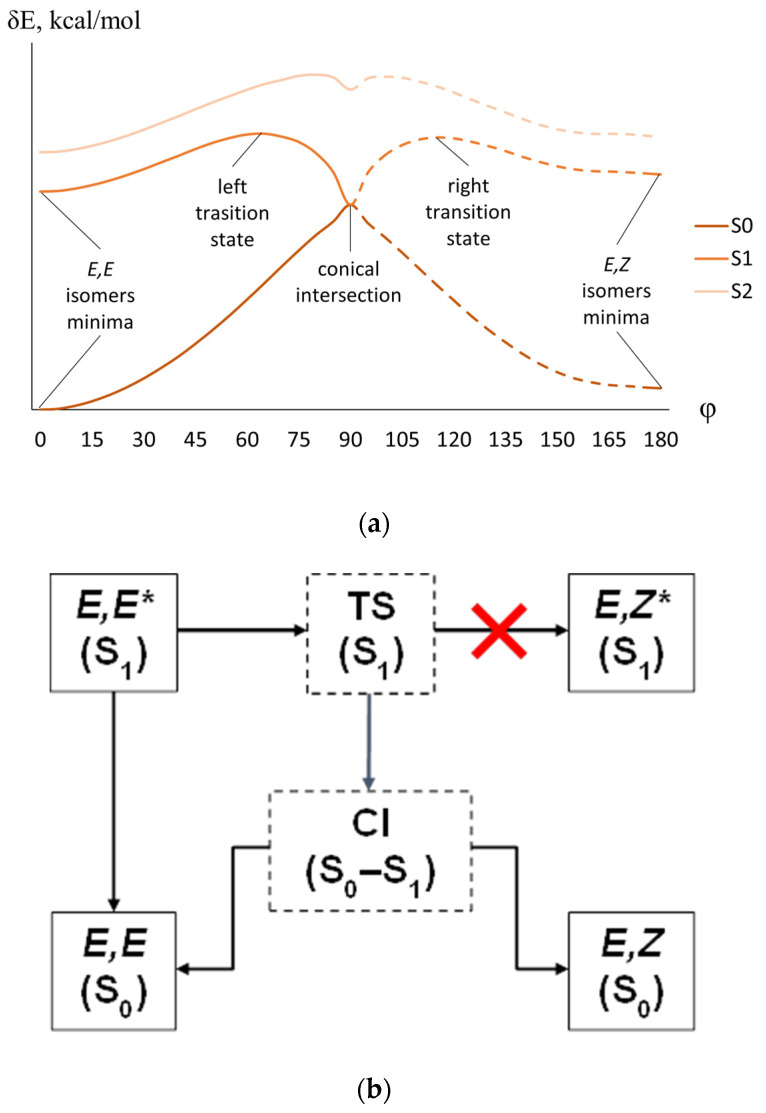
(**a**) Typical potential energy profile of the ground S0 and lowest excited S1 (ππ*) and S2 (nπ*) states. (**b**) Relaxation processes resulting in *trans–cis* isomerization.

**Figure 7 ijms-24-13362-f007:**
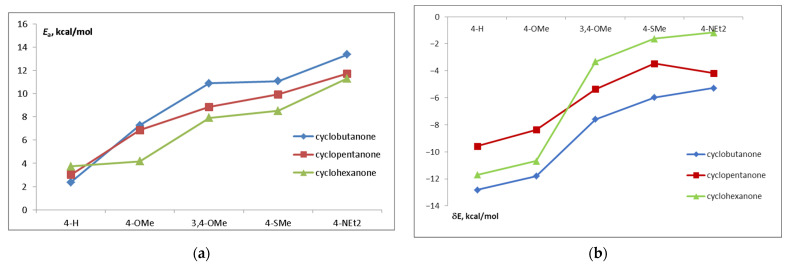
(**a**) The activation energy for the transition from (*E*,*E*) to CI and (**b**) CI depth in the S1 state.

**Table 1 ijms-24-13362-t001:** Torsion angle C2-C7-C16-C21 (°, shown by arrow in Figure 2) characterizing sterical distortion of the π system.

	a	b	c	d	e
**1**	0.1	0.0	0.0	0.0	0.4
**2**	7.0	1.0	0.1	0.5	0.2
**3**	30.3	26.9	23.9	26.6	21.0

**Table 2 ijms-24-13362-t002:** Mulliken charges on donor fragments.

	a	b	c	d	e
**1**	0.11	0.14	0.14	0.13	0.20
**2**	0.10	0.13	0.12	0.11	0.19
**3**	0.07	0.10	0.10	0.09	0.16

**Table 3 ijms-24-13362-t003:** Calculated radiative lifetime (τ_r_) of *E*,*E* isomer of cycloalkanones, (*E*,*E*)–(*E*,*Z*) isomerization rate constant (*k*), and isomerization time (t_tc_).

Dienone	τ_r_, ns	*k*, s^−1^	t_tc_, ns
**Cyclobutanone**			
**1a**	-	1.12 × 10^10^	0.09
**1b**	2.28	4.97 × 10^6^	201
**1c**	3.05	4.40 × 10^3^	2.27 × 10^5^
**1d**	2.62	4.06 × 10^3^	2.465 × 10^6^
**1e**	3.07	1.22 × 10^2^	8.183 × 10^6^
**Cyclopentanone**			
**2a**	-	4.66 × 10^9^	0.21
**2b**	1.83	8.28 × 10^6^	121
**2c**	2.23	8.13 × 10^4^	1.23 × 10^4^
**2d**	1.97	3.34 × 10^4^	2.99 × 10^4^
**2e**	1.90	2.55 × 10^4^	3.92 × 10^4^
**Cyclohexanone**			
**3a**	-	1.41 × 10^9^	0.71
**3b**	2.90	5.47 × 10^8^	1.83
**3c**	2.26	7.12 × 10^5^	1404
**3d**	2.02	4.92 × 10^5^	2033
**3e**	2.46	3.40 × 10^3^	2.94 × 10^5^

## Data Availability

The data presented in this study are available on request from the corresponding author.

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
