# Peer review of "Theoretical Study of Structure and Photophysics of Homologous Series of Bis(arylydene)cycloalkanones"

_ijms, 2023, doi:10.3390/ijms241713362_

Round 1
Reviewer 1 Report
The results obtained by the authors in this manuscript are very interesting, as are the results in all other works on the study of photophysical properties.
I recommend this work for publication with some remarks:
1. In the appendix (or in the text) it is necessary to indicate which theoretical level was used to calculate the NMR chemical shifts, as well as the experimental and calculated NMR standard.
2. The authors found correlations between the calculated HOMO and LUMO energies and the experimental oxidation and reduction potentials, as well as between the calculated HOMO-LUMO gap and the experimental gap between the oxidation and reduction potentials. The R2 values are quite high. Can the authors say something about the physical meaning of the constants obtained for linear dependencies?
Author Response
We would like to thank the reviewer for their kind remarks.
- In the appendix (or in the text) it is necessary to indicate which theoretical level was used to calculate the NMR chemical shifts, as well as the experimental and calculated NMR standard.
1H NMR spectra were calculated relative to TMS as a standard using the Priroda program package with the PBE functional and triple-ζ quality basis set 3z implemented in the program. The optimized geometries were taken from the PBE0/6-31+G(d,p)/DPCM calculation.
The experimental spectra were also recorded relative to TMS (see corresponding experimental papers, Refs. [1-3])
- The authors found correlations between the calculated HOMO and LUMO energies and the experimental oxidation and reduction potentials, as well as between the calculated HOMO-LUMO gap and the experimental gap between the oxidation and reduction potentials.The R2 values are quite high. Can the authors say something about the physical meaning of the constants obtained for linear dependencies?
We added the following paragraphs to the manuscript:
In the Hartree–Fock theory, orbital energies of the occupied orbitals are negative of the corresponding ionization potentials (IPs), and orbital energies of unoccupied orbitals are negative of the electron affinities (EAs). This relationship is called Koopmans theorem. In the Density Functional Theory, however, this is only valid for the HOMO and first IP. Nevertheless, some correlation can be observed between the LUMO energy and first EA. In addition, HOMO-LUMO gap in DFT correlates with the energy of the first electronic transition. At the same time, IP and EA are related to the oxidation and reduction potentials. Table S2 of Supplementary Materials shows the calculated IPs and EAs compared to the experimental Eox and Ered. The correlation is also very good.
In the case of the perfect agreement between the calculated and experimental values, the linear regression coefficient should be close to 1. Our data shows that for the HOMO, IP, and Eox, this coefficient is more than 0.9, which indicates very good agreement. For the LUMO, EA, and Ered, this coefficient is ~0.7-0.8 with very high R2 value. This is not surprising, because the energies of virtual orbitals both in Hartree–Fock and DFT are calculated with less reliability. To some extent, it may be remedied through the use of optimally tuned range-separated functionals. The intercept on the linear regressions of Eox(Ered) vs. HOMO(LUMO) or vs. IP(EA) incorporates all the systematic errors and the absolute potential of the reference electrode.
Reviewer 2 Report
In this work, Starostin et al. employed a computational approach to provide information of structure and photophysics of series of bis(arylidene)cycloalkanones dyes with various donor substituents. The search for new dyes with specific photophysical properties is an extremely important, but also expensive topic. Therefore, if the relationship between the structure and luminescent properties of compounds can be predicted computationally, it is helpful in the design of new systems. So the work fits the current trends in photochemistry well. The work presents interesting research results. It is also relatively well written with some minor shortcomings. The calculation have been done carefully, but some points should be clarified. My opinion is that the topic is of interest to the readership of International Journal of Molecular Sciences so the manuscript is publishable, but after major revision:
1. It would be interesting to comment on the choice of the functional PBE0 compared to other functional ones. Why did the authors decide to choose this functional for calculations?
2. In the Materials and Methods section does not provide references to either the functional or the basis set. In addition, on page 2 (in the Results section) no reference to the PCM solvent model is provided.
3. Throughout the article, the authors give a lot of experimental data (H NMR shifts, Eox-Ered gap, UV-Vis and emission spectra) that do not contain any references. This should be completed.
4. In the graphs both in the article (Figure 3) and in supplementary materials (Figure S1), the axes are not signed.
5. Torsion angles are given in Table 1. Between which atoms? Wouldn't it be more precise to measure the angles between planes (eg. between planes formed by benzene rings).
6. In the Materials and Methods section, was given how the ionization potential and electron affinities were calculated. However, the results of these calculations are not given in the entire article. Why?
The article needs a minor linguistic correction.
Author Response
We would like to thank the reviewer for their valuable comments.
- It would be interesting to comment on the choice of the functional PBE0 compared to other functional ones. Why did the authors decide to choose this functional for calculations?
PBE0 was used in our previous calculations of these and similar dyes. The electronic transitions in the D-A and D-A-D type dyes show noticeable charge transfer character; therefore, some fraction of exact exchange is needed to avoid artifacts (underestimated energies of charge-transfer states with zero oscillator strength). At the same time, exact exchange in the functional leads to overestimation of the energies of non-artifact states both with nonzero and zero oscillator strength. 20-25% of HF exchange allow one to get rid of the artifact states (by making their energy high enough) and cause only moderate overestimation of the energies of the target states. Therefore, we chose PBE0 (25% HF exchange).
- In theMaterials and Methodssection does not provide references to either the functional or the basis set. In addition, on page 2 (in the Results section) no reference to the PCM solvent model is provided.
All the calculations were performed with PBE0/6-31+G(d,p). This is mentioned in the first paragraph of the Materials and Methods section. Only the chemical shifts were calculated with PBE/3z (triple-zeta quality basis set implemented in Priroda package) using the geometries optimized previously with PBE0/6-31+G(d,p)/DPCM. We are sorry for not mentioning this in the Materials and Methods section.
- Throughout the article, the authors give a lot of experimental data (H NMR shifts, Eox-Ered gap, UV-Vis and emission spectra) that do not contain any references. This should be completed.
All the experimental data are from the following papers:
[1] Fomina, M.V.; Vatsadze, S.Z.; Freidzon, A.Y.; Kuz’mina, L.G.; Moiseeva, A.A.; Starostin, R.O.; Nuriev, V.N.; Gromov, S.P. Structure–Property Relationships of dibenzylidenecyclohexanones. ACS Omega 2022, 7, 10087–10099.
[2] Fomina, M.V.; Freidzon, A.Y.; Kuz’mina, L.G.; Moiseeva, A.A.; Starostin, R.O.; Kurchavov, N.A.; Nuriev, V.N.; Gromov, S.P. Synthesis, Structure and Photochemistry of Dibenzylidenecyclobutanones. Molecules 2022, 27, 7602.
[3] Vatsadze, S.Z.; Gavrilova, G.V.; Zyuz’kevich, F.S.; Nuriev, V.N.; Krut’ko, D.P.; Moiseeva, A.A.; Shumyantsev, A.V.; Vedernikov, A.I.; Churakov, A.V.; Kuz’mina, L.G.; et al. Synthesis, structure, electrochemistry, and photophysics of 2,5- dibenzylidenecyclopentanones containing in benzene rings substituents different in polarity. Russ. Chem. Bull. 2016, 65, 1761–1772.
We added the corresponding references to the text and the supplementary materials.
- In the graphs both in the article (Figure 3) and in supplementary materials (Figure S1), the axes are not signed.
Corrected
- Torsion angles are given in Table 1. Between which atoms? Wouldn't it be more precise to measure the angles between planes (eg. between planes formed by benzene rings).
The torsion angles are shown by the arrows in Fig. 1. In the revised version, the corresponding atom numbers are added.
- In theMaterials and Methodssection, was given how the ionization potential and electron affinities were calculated. However, the results of these calculations are not given in the entire article. Why?
We added Table S2 to supplementary materials.
Round 2
Reviewer 2 Report
The Authors addressed all my comment, so the decision on acceptance the manuscript can be made.